# The Adipose Stem Cell as a Novel Metabolic Actor in Adrenocortical Carcinoma Progression: Evidence from an In Vitro Tumor Microenvironment Crosstalk Model

**DOI:** 10.3390/cancers11121931

**Published:** 2019-12-04

**Authors:** Roberta Armignacco, Giulia Cantini, Giada Poli, Daniele Guasti, Gabriella Nesi, Paolo Romagnoli, Massimo Mannelli, Michaela Luconi

**Affiliations:** 1Endocrinology Unit, Department of Experimental and Clinical Biomedical Sciences “Mario Serio”, University of Florence, 50139 Florence, Italy; giulia.cantini@unifi.it (G.C.); giadapoli@tiscali.it (G.P.); massimo.mannelli@unifi.it (M.M.); 2Histology & Embryology Unit, Department of Experimental and Clinical Medicine, University of Florence, 50139 Florence, Italy; daniele.guasti@unifi.it (D.G.); paolo.romagnoli@unifi.it (P.R.); 3Division of Pathological Anatomy, Department of Experimental and Clinical Medicine, University of Florence, 50139 Florence, Italy; gabriella.nesi@unifi.it; 4Istituto Nazionale Biostrutture e Biosistemi (INBB), viale delle Medaglie d’Oro 305, 00136 Rome, Italy

**Keywords:** adipose precursors, adipogenesis, invasion, cancer, cell reprogramming, leptin

## Abstract

Metabolic interplay between the tumor microenvironment and cancer cells is a potential target for novel anti-cancer approaches. Among stromal components, adipocytes and adipose precursors have been shown to actively participate in tumor progression in several solid malignancies. In adrenocortical carcinoma (ACC), a rare endocrine neoplasia with a poor prognosis, cancer cells often infiltrate the fat mass surrounding the adrenal organ, enabling possible crosstalk with the adipose cells. Here, by using an in vitro co-culture system, we show that the interaction between adipose-derived stem cells (ASCs) and the adrenocortical cancer cell line H295R leads to metabolic and functional reprogramming of both cell types: cancer cells limit differentiation and increase proliferation of ASCs, which in turn support tumor growth and invasion. This effect associates with a shift from the paracrine cancer-promoting IGF2 axis towards an ASC-associated leptin axis, along with a shift in the SDF-1 axis towards CXCR7 expression in H295R cells. In conclusion, our findings suggest that adipose precursors, as pivotal components of the ACC microenvironment, promote cancer cell reprogramming and invasion, opening new perspectives for the development of more effective therapeutic approaches.

## 1. Introduction

A growing tumor mass can be considered as a dynamic “pseudo-organ” in which active molecular and metabolic interaction is established between cancer cells and different populations of stromal, vascular and immune cells, constituting the complex cell network of the tumor microenvironment (TME) that modulates and is reciprocally modulated by the cancer mass [1]. These “companion” cells are pivotal in favoring cancer cell adaptation to the different challenging conditions associated with tumor progression, growth and invasion, thus representing a potential novel anti-tumor target. Several studies have demonstrated that tumor cells can recruit fibroblasts and mesenchymal cells at the lesion site, modifying their properties and metabolism to be functional to tumor evolution [2,3]. In recent years, adipocytes have raised great attention as active players in the TME [4]. Far from being mere passive bystanders, adipocytes can store and release energy in the form of fatty acids and secrete specific adipose regulatory factors, namely adipokines. They thus mediate both local and systemic effects, with particular implications for tumor initiation and growth, as well as local invasion and metastasis [4]. Moreover, increasing evidence associates obesity (i.e., adipose tissue expansion/dysfunction and an altered balance between energy intake and consumption) with a higher risk of cancer development, progression and mortality [5,6,7,8].

In many solid tumors occurring in organs with particularly abundant fat, the surrounding adipose tissue is often invaded by cancer cells, thus enabling close contact with adipocytes at the tumor invasive front [9,10,11,12]. Invading cells are indeed able to actively interact with the surrounding adipocytes and deeply modify both their phenotype and metabolism. Conversion towards cancer-associated adipocytes (CAA) has been described, characterized by de-lipidation/de-differentiation, free fatty acid release and immunomodulatory adipokine secretion, as well as a shift in the OXPHOS metabolism towards glycolysis, resulting in the so-called reverse-Warburg effect [10,11,13,14,15,16,17]. This re-programming of adipose cells provides the cancer cells with the substrates to fuel β-oxidation and the oncometabolites necessary for growth and invasion. In addition, adipose-derived stem cells (ASCs) seem capable of acquiring properties similar to those of cancer associated-fibroblasts in the TME [18] and promoting cancer cell proliferation, viability and invasiveness, as well as chemoresistance [19,20,21]. Nevertheless, the factors and mechanisms involved in this interaction still need to be clarified.

Adrenocortical carcinoma (ACC) is a rare endocrine malignancy affecting the adrenal cortex, has a high risk of relapse and metastasis, and is often resistant to the current standard-of-care medical therapies, consisting of radical surgery and mitotane administration. ACC prognosis is usually poor, especially when the tumor is metastatic at diagnosis. Recent advances in the genetic characterization of ACC may open therapies toward the development of more specific and effective treatments [22]. However, it is essential to clarify the mechanisms underlying the aggressive phenotype in ACC, which still remain elusive. Exploring the molecular interaction with the TME may help dissecting the mechanisms promoting tumor progression and may provide potential new targets for therapy.

Physiologically, adrenal glands are surrounded by a huge depot of visceral adipose tissue, in close contact with the tumor when the tumor mass develops and the adrenal capsule disrupts. Therefore, ACC is a good model to assess the effect of crosstalk occurring between cancer cells and adipose precursors. In this paper, we investigated the effects of the interaction between primary human ASCs and the ACC cell line H295R in an in vitro indirect co-culture system.

## 2. Results

### 2.1. Proliferative, Metabolic and Morphological Changes in ASCs Co-Cultured with H295R Cells

Advanced ACC is often characterized by capsular invasion with infiltration of adjacent tissues, particularly the visceral fat depots surrounding the adrenal gland. In our cohort of advanced ACC patients (stage 3: *n* = 15, stage 4: *n* = 4), capsular invasion is present in 89% of tumors (17/19) (Figure 1).

In this context, a close contact between adrenocortical cancer cells and cells of the adipose lineage (adipose precursors and differentiated adipocytes) extensively occurs. We tried to reproduce this microenvironment interaction by setting up an indirect in vitro co-culture system between the adrenocortical cancer cell line H295R and primary cultures of adipose stem cells derived from adipose tissue specimens [23,24]. By using a system in which the two cell types were cultured together but physically separated by membrane permeable to soluble factors, we evaluated the putative crosstalk established between the two compartments under different conditions.

We first focused on the effect of the co-culture system on the adipose stem cell behavior and functions. Human ASCs were co-cultured with H295R cells up to 9 days. We observed a statistically significant increase in the proliferative rate of the co-cultured ASCs, compared with the ASC mono-culture, starting from day 7 and reaching a maximum at day 9 (3.8 ± 0.3-fold and 10.1 ± 1.7-fold, respectively) (Figure 2A).

This increased proliferation was accompanied by a significant increase in glucose uptake measured at day 7 of co-culture (2.06 ± 0.11-fold) (Figure 2B) and, consistently, by the up-regulated expression of insulin-independent glucose transporter-1 (GLUT-1), but not of the insulin-dependent form GLUT-4, as assessed by western blot analysis (Figure 2B, inset). Glucose and lactate concentrations in the ASC-conditioned medium were also measured in order to evaluate any metabolic switch toward aerobic glycolysis. In co-culture conditions, we measured decreased levels of glucose compared with the mono-culture, consistently with the observed increase in glucose uptake; conversely, both extracellular and intracellular lactate levels were significantly increased (Table 1), suggesting that the boosted ASC proliferation may be preferentially fueled by aerobic glycolysis.

In line with their mesenchymal stem origin, ASCs expressed a set of specific markers, including Bmi1, Nanog and OCT4, which associate with the stem potential maintenance. Following co-culture with H295R cells, the expression of all these stem genes was significantly decreased compared to control ASCs (Figure 2C). 

Consistent with the reduced stem potential and a likely induced differentiation toward a fibroblast-like phenotype, a significant increase in the myofibroblast-like marker alpha-smooth muscle actin (α-SMA) was detected by western blot analysis in ASCs cultured for 7 days with H295R cells (Figure 2D). An associated macroscopic change in the cell morphology was observed: co-cultured ASCs showed a more elongated and fibroblastoid-like shape and displayed the presence of bundles compared with the controls (Figure 2D).

To better appreciate the different morphology at the ultrastructural level, we performed transmission electron microscopy of ASCs after 7-day culture. When cultured alone, ASCs showed wide portions of plasma membrane organized in microvilli, and cytoplasm rich in dense lysosome-like bodies. Mitochondria appeared elongated (length > 1 μm) and often branched, with typical transverse cristae. Rough endoplasmic reticulum and Golgi apparatus were poorly developed. Small cytoplasmic vesicles were also present, presumably endosomes or lysosome precursors (Figure 3A,B).

After co-culture, ASCs showed cytoplasmic features of active protein secreting cells: the rough endoplasmic reticulum was expanded, with several ribosomes associated to its surface; Golgi cisternae were increased in number and size and lysosome-like bodies were significantly reduced. Moreover, under co-culture conditions we observed the presence of several small lipid droplets (Figure 3C,E), suggestive of limited differentiation towards adipocytes, and an increased number of smaller and roundish mitochondria (Figure 3D).

### 2.2. H295R Cells Hamper ASC Ability to Differentiate toward Mature White Adipocytes

To assess any effect of H295R-co-culturing on adipogenic potential, ASCs maintained in mono- or co-culture were induced to differentiate in vitro towards mature adipocytes. Quantitative RT-PCR Taqman analysis (RT-qPCR) revealed a drastically reduced expression of three genes specifically expressed by mature adipocytes, namely Adiponectin (AdipoQ), Fatty Acid Binding Protein 4 (FABP4) and Hormone Sensitive Lipase (HSL), in adipose cells induced to differentiate during co-culture with the tumor cells (Figure 4A).

As the level of adipocyte differentiation positively correlated with the amount of intracellular lipids stored in triglyceride droplets, we also evaluated the lipid content of differentiated ASCs. The quantification of cell triglyceride content by AdipoRed assay showed that adipocytes obtained during co-culture had a significantly decreased intracellular lipid content compared with adipocytes obtained in mono-culture (Figure 4B), thus indicating that the presence of cancer cells affects the ability of ASCs to efficiently differentiate toward mature adipocytes. Consistent, epifluorescence microscopy images of Adipored-stained adipocytes revealed a substantial difference in the lipid droplet size of adipocytes in mono and co-culture: ASCs cultured alone differentiated in adipocytes with larger lipid droplets stuffing the entire cytoplasm, while adipocytes derived from co-cultured ASCs show less and smaller lipid droplets (Figure 4C).

### 2.3. H295R Cells Increase Proliferation and Glucose Metabolism When Co-Cultured with ASCs

In the same co-culture system setting described above, we further evaluated the co-culture effect on the tumor cell counterpart. As it had been observed for the adipose precursors, H295R cells showed a significant increase in cell proliferation at day 9 co-culture, compared with H295R cells cultured alone (Figure 5A).

H295R cells in co-culture also showed an increased glucose uptake compared with mono-culture, as evaluated by 2-deoxy-[^3^H]D-glucose cell incorporation after 9 days (Figure 5B). Consistently, lower glucose and higher lactate concentrations were measured in the conditioned medium of co-cultured H295R cells compared with the control, as well as an increased intracellular lactate content (Table 1). In order to investigate any involvement of the IGF-1R/ERK axis, which sustains the proliferative autocrine loop in adrenal cancer [25], we performed western blot analysis on lysates of H295R cells cultured alone or with ASCs for 7 days. A significant increase in MEK expression and ERK phosphorylation was observed in co-culture conditions, indicating an increased activation of the ERK signaling (Figure 5C). Transmission electron microscopy performed on H295R cells showed a sustained decrease in intracellular lipid droplets when H295R cells were cultured in the presence of ASCs compared with the mono-cultures (Figure 5D).

### 2.4. H295R Increase Migration and Invasive Ability after Co-Culture with ASCs

Beyond cell proliferation, we also assessed whether the presence of ASCs could improve cancer cell motility and invasive potential. H295R cell ability to heal an induced wound was assessed by scratch test after 7-day co-culture with ASCs. The percentage of migrating cells able to cover the scratch was significantly higher if H295R cells were previously co-cultured with ASCs compared with the control (Figure 6A). Co-cultured H295R cells also showed an increased invasive ability compared with H295R cultured alone (Figure 6B).

To further support the evidence that ASC co-culture increased H295R invasiveness potential, we assessed the ability of the latter cells to migrate through an endothelial monolayer by using a trans-endothelial migration assay. Co-cultured H295R cells exhibited a statistically significant higher ability to cross the endothelial barrier consisting of a human microvascular endothelial cell (HMEC-1) monolayer, compared with H295R cells cultured alone (Figure 6C). In addition, by performing immunofluorescence staining for F-actin, we evaluated the remodeling of the actin cytoskeleton organization: cancer cells cultured in the presence of ASCs showed an altered morphology, acquiring a more elongated and polarized shape. Particularly, cytoplasmic stress fibers appeared less defined and F-actin formed focal structures rather than being homogeneously distributed along the plasma membrane (Figure 6D). This suggests that a cytoskeletal re-organization supporting cancer cell migration was induced during co-culture. Consistently, protein expression of Focal Adhesion Kinase (FAK), RhoA and, to a lesser extent, Fascin-1, all involved in the formation of lamellipodia, filopodia and focal adhesions essential for cell migration and invasion, was increased in co-cultured H295R cells compared with the control (Figure 6E).

### 2.5. Molecular Mechanisms Activated by ASCs/H295R Cell Crosstalk

Since the indirect co-culture system described above mimics a paracrine cell interaction, we further investigated which soluble factors may be specifically produced by ASCs and potentially involved in the tumor progression. Among the adipokines produced by ASCs, leptin and interleukin-8 (IL-8), have been shown to mediate cancer proliferation, migration and invasion [11,26,27,28,29]. Accordingly, gene expression analysis of co-cultured ASCs revealed a statistically significant increment of both leptin and IL-8 compared with ASC mono-culture (Figure 7A). A concomitant increase in leptin receptor (Ob-R) and a decreased expression of both IGF2 and its receptor IGF-1R were observed in H295R cells when co-cultured with ASCs (Figure 7B).

To elucidate the molecular mechanisms underlying the higher motility/migratory ability observed in H295R cells after co-culture, we investigated the modulation of the stromal cell-derived factor 1 (SDF-1 or CXCL12)/CXCR4/CXCR7 axis in the different culturing conditions. In fact, SDF-1 is among the main chemokines produced by ASCs and mesenchymal cells and its paracrine signaling contributes to tumor proliferation, migration and invasion [30]. Although SDF-1 gene and protein expression were reduced in lysates of co-cultured ASCs (Figure 7C), we detected significantly higher levels of the secreted form of SDF-1 in the conditioned medium of co-cultured ASCs compared with the control (Figure 7D). In addition, we observed a concomitant decrease in gene expression of the Dipeptidyl Peptidase 4 (DPP4) enzyme, which is a peptidase able to degrade multiple chemokines, including SDF-1. Therefore, higher levels of SDF-1 in the conditioned media seemed to be maintained at a steady state by reduced degradation instead of increased production by ASCs. Finally, when the expression of the main SDF-1 receptors, CXCR4 and CXCR7, was assessed, we observed a decrease in CXCR4 and a concomitant increase in CXCR7 expression in co-cultured H295R cells compared with mono-culture control (Figure 7E).

### 2.6. A Reciprocal Interaction Is Also Established between H295R Cells and Mature Adipocytes

Since the tumor adipose microenvironment mainly consists of mature adipocytes with their stem pool of ASCs, the effect of co-culture between H295R cells and in vitro differentiated mature adipocytes was also assessed. The proliferative rate of H295R cells co-cultured with mature adipocytes was slightly but not significantly increased compared with H295R alone (5.01 ± 0.28 vs. 4.68 ± 0.86-fold, *n* = 3 independent experiments). Western blot analysis revealed a higher expression of MEK and p-ERK (Figure 8), indicating that the MAPK-mediated pathway is enhanced by the presence of adipocytes. Moreover, the expression of the migration-related proteins FAK, RhoA and Fascin-1 was also increased (Figure 8), suggesting that mature adipocytes can also contribute to modulating cancer cell progression by reprogramming cancer cells signaling. 

We further evaluated the reciprocal effect of H295R cells on co-cultured mature adipocytes by assessing AdipoQ and FABP4 gene expression levels. A statistically significant decrease in the expression of both genes was observed in mature adipocytes co-cultured with H295R cells compared with adipocytes alone (Figure 9A). A significant decrease in adipocyte intracellular lipid content was also observed in co-cultured adipocytes vs. mono-culture, as quantified by AdipoRed assay (Figure 9B) and confirmed by epifluorescence microscopy (Figure 9C–E), suggesting a de-lipidation effect. Finally, adipocytes co-cultured with H295R cells expressed significantly higher levels of leptin compared with the mono-culture, as quantified by RT-qPCR (4.50 ± 0.3-fold, *p* < 0.001, *n* = 5 independent experiments).

## 3. Discussion

Despite the recent insights into the molecular characterization of ACC, the mechanisms underlying tumor progression and aggressiveness still need to be elucidated [31]. Accordingly, any modern approach can no longer focus on tumor cells alone and therefore we directed our study to include the plastic interacting network of the surrounding adipose microenvironment. Indeed, the emerging mechanisms involving mature adipocytes and lipid metabolism have been reported to play a role in reprogramming cancer cells and altering the response to cancer treatment [32].

In this paper, we demonstrated that H295R cells and the adipose precursors reciprocally stimulated each other to proliferate. Moreover, the tumor cells restrained ASC adipogenic potential, recruiting the stem cells to committed, but not fully differentiated, pre-adipocytes. In turn, the adipose microenvironment contributed to re-programming H295R cells toward a more invasive phenotype, by changing their metabolic requirements and dependence from the IGF2 paracrine axis to the ASC-dependent leptin production, and inducing a shift in the expression of chemokine receptors involved in supporting tumor invasion from CXCR4 to CXCR7. Further, we showed that a reciprocal interaction also occurs between H295R cells and co-cultured mature adipocytes.

Indeed, it is now widely recognized that the tumor microenvironment (TME) plays a pivotal role in sustaining tumor metabolic adaptation to challenging conditions, allowing cancer cells to locally invade and eventually metastasize. Mesenchymal stem cells (MSCs), which in the normal tissue are involved in the processes of homeostasis, repair and regeneration, act as key players at distinct steps of tumorigenesis [3]. The adipose tissue is a rich source of MSCs and it is a physiologically abundant component of the adrenal gland. Particularly in advanced ACC, where the tumor disrupts the adrenal capsule and invades the surrounding fat, this visceral adipose depot is in close contact with the tumor cells at the tumor mass periphery. Indeed, we reported that in our cohort of patients with advanced ACC (stage 3–4), the disruption of the adrenal capsule occur in 89% of the cases, exposing cancer cells to a close contact with the adrenal surrounding adipose tissue. Thus, ASC recruitment can occur, with a consequent establishment of a bi-directional interplay with the cancer cells. This hypothesis is strongly supported by our findings in the in vitro model of co-culture between ASCs and the adrenocortical cancer cell line H295R, where phenotypic and metabolic changes were observed in both cell types. ASCs exhibited a huge remodeling under co-culture conditions as shown by light microscopy and alpha-SMA expression: they increased their proliferation rate and acquired a phenotype half way between myofibroblasts and poorly differentiated adipose cells/pre-adipocytes, as shown by electron microscopy, and reported in other tumor types [33,34,35,36]. Metabolic cancer cell-induced reprogramming in ASCs was confirmed by the observed modifications of the intracellular ultra-structures, with the presence of lipid droplets, an expanded protein secretion machinery and a different mitochondria shape, which particularly associates with changes in cell metabolism [37]. The concept that cancer cells induce a less differentiated and more plastic phenotype in adipose precursors, useful for the tumor, was further supported in our system by the observation that ASCs showed an impaired ability to give rise to mature adipocytes when induced to differentiate in the presence of H295R cells. This reduced maturation level was characterized by a lower expression of markers of differentiated adipocytes, such as FABP4 and adiponectin. Moreover, co-culture experiments with mature adipocytes also resulted in a decrease in adipogenic differentiation markers and of intracellular lipid content. These phenomena are consistent with the de-differentiation/de-lipidation process observed for other tumor types, where mature adipocytes around the cancer mass are induced to loose intracellular lipid droplets and regress to fibroblastoid precursors, finally transforming into the so called cancer-associated adipocytes (CAAs), that sustain tumor functions [13,14,16].

The metabolic reprogramming required for tumor adaptation, proliferation and malignant progression generally relies on the acquisition of the Warburg phenotype, due to the metabolic switch of cells from the OXPHOS respiration towards the aerobic glycolysis [38]. This is likely the case in our co-culture system, where both ASCs and H295R cells displayed increased glucose uptake and higher intracellular lactate content, with a concomitant fall in glucose and rise in lactic acid concentrations in the conditioned media. This metabolic switch could explain the increased proliferation rate of both cell types under co-culture conditions.

Co-culture conditions are also characterized by a decrease in cancer cell intracellular lipid droplets. Aggressive ACC has been associated with a molecular signature characterized by hyper-methylation/inactivated expression of the G0/G1 switch gene 2 (G0S2) [39,40]. Interestingly, this gene normally enhances intracellular triglyceride accumulation and inhibits lipolysis in adipocytes, suggesting that the repressed expression described in advanced/invasive ACC may be associated with reduced intracellular lipid droplets in cancer cells, similarly to the phenomenon observed in H295R after co-culturing with adipose stem cells. Nevertheless, further studies are required to assess the involvement of an increased hyper-methylation of G0S2 in H295R cell in co-culture, linking reduced lipid storage to the more invasive phenotype.

Moreover, intracellular accumulation of lactate, considered as a master oncometabolite involved in many of the main steps of tumor progression [41], may also sustain acquisition of a more invasive phenotype by H295R cells, as we observed in co-culture. This increase in motility and migration ability may derive from activation of specific molecular pathways by signaling molecule exchange within the system.

Circulating leptin is one of the factors involved in the association between obesity and some solid cancers [42]. Leptin, locally secreted in the TME by both adipocytes and cancer-associated fibroblasts (CAFs) [43], has been shown to be involved in various steps of tumor invasion and metastasis through interaction with its Ob-R receptor expressed by cancer cells and activation of the downstream signaling pathway JAK-STAT3, PI3K/AKT and MAPK [29]. Consistently, when ASCs and H295R cells were co-cultured, the increased expression of leptin and Ob-R in adipose and tumor cells respectively, might suggest the activation of a specific cancer promoting axis. The increased expression of MEK and phosphorylated ERK and the concomitant decrease in IGF2 and IGF-1R expression in co-culture conditions might also indicate a shift from the endogenous paracrine IGF2 axis towards a cancer promoting ASC-associated leptin axis. Among the downstream effectors activated by the Ob-R-associated signaling, FAK and RhoA, two molecules involved in the cell-to-extracellular matrix adhesion necessary for migratory cell movement, were up-regulated in co-cultured H295R cells, as well as in co-culture with mature adipocytes. Leptin promotes EMT and invasion also by upregulating IL-8 production in the TME [11,44,45]. Consistently, we found a significant increase in the expression of IL-8 in ASCs when co-cultured with H295R cells, suggesting that the IL-8 production may contribute to leptin promotion of the aggressive phenotype in H295R cells. In addition, we observed an increased protein expression of Fascin-1, an actin-binding protein participating in the organization and functionality of cell protrusions, recently shown to associate with ACC invasiveness [46].

The CXCL12 (SDF-1)/CXCR4/CXCR7 axis, which has been shown to contribute to tumor progression by modulating cancer cell survival, proliferation and migration [30], also seemed to be involved in the ASC/H295R cell crosstalk. Higher levels of SDF-1 acting on H295R cells were maintained in the co-culture conditioned media by a reduced cleavage by DDP4 rather than by an increased production from ASCs. The decreased expression of CXCR4 and the concomitant increase in CXCR7 in H295R cells, in response to the increased level of SDF-1 measured in ASC conditioned medium, suggests a preferential activation of the CXCL12/CXCR7-related signaling in this co-culture system. A similar CXCR7 over-expression has been described in a variety of cancers, with particular implication for cancer initiation, progression and metastasis [47,48,49,50,51,52,53], mainly mediated via ERK activation [54,55]. In addition, CXCR7 seems essential for the process of trans-endothelial migration, that is a pivotal step for supporting tumor cell entrance in the bloodstream and metastasis [56]. Indeed, under co-culture conditions the shift in the CXCR4/CXCR7 ratio towards the CXCR7 axis was associated not only with increased invasion potential but also with increased ability of H295R cells to migrate through an endothelial monolayer system.

## 4. Materials and Methods

### 4.1. Reagents

Primary antibodies against Fascin-1 [mouse monoclonal (D-10) sc-46675], RhoA [mouse monoclonal (26C4) sc-418], GAPDH [rabbit polyclonal (FL-335) sc-25778], Actin [goat polyclonal (C-11) sc-1615], GLUT1 [mouse monoclonal (A-4) sc-377228], GLUT4 [rabbit polyclonal (H-61) sc-7938] and SDF-1/CXCL12 [rabbit polyclonal (FL-93) sc-28876] were from Santa Cruz Biotechnology, Inc. (Dallas, TX, USA); anti-phospho-p44/42 ERK (Thr202/Tyr204) [mouse monoclonal, #9106S] and p44/42 ERK1/2 [rabbit polyclonal, #9102] from Cell Signaling Technology, Inc. (Danvers, MA, USA); anti-FAK [rabbit polyclonal, GTX132141] was from GeneTex, Inc. (Irvine, CA, USA); anti-MEK1[rabbit monoclonal, #04-376] was from EMD-Millipore (Burlington, MA, USA); anti-α-smooth muscle actin (mouse monoclonal (1A4), #A2547) was from Sigma-Aldrich (Saint Louis, MO, USA). Rabbit (#A0545) and goat (#A4174) peroxidase-conjugated secondary antibodies, media and sera for cell cultures were purchased from Sigma-Aldrich; mouse peroxidase-conjugated secondary antibody (#sc-2005) was from Santa Cruz Biotechnology, Inc. Plasticware was obtained from Corning Inc. (Corning, NY, USA). 2-deoxy-[3H]-D-glucose was provided by Perkin Elmer (Waltham, MA, USA). Other reagents for cell culture and microscopy were obtained from Sigma-Aldrich, except where otherwise indicated**.**

### 4.2. Histology of ACC

Histologic diagnosis of ACC was recorded by a reference pathologist (G.N.) on tumor tissue removed at surgery. Four-micrometer-thick sections were cut from formalin fixed, paraffin-embedded tissue blocks and stained with hematoxylin and eosin. Tumor specimens were evaluated according to the Weiss score system in which the presence of three or more criteria highly correlates with malignant behavior [57]. Tumor stage was evaluated according to the revised TNM classification of ACC proposed by the European Network for the Study of Adrenal Tumors [58]. The Weiss score parameter of capsular invasion was reported in this study from our cohort of patients with ACC stage 3–4 (*n* = 15, stage 3 and *n* = 4, stage 4) [46,59].

### 4.3. Cell Culture

The human ACC cell line H295R was obtained from American Type Culture Collection (Manassas, VA, USA) and cultured in DMEM/F-12 medium with 10% FBS, 2 mM L-glutamine, 100 U/mL penicillin, 100 μg/mL streptomycin, enriched with a mixture of insulin/transferrin/selenium. H295R cell line has been used for experiments between passages 16–21.

Human primary adipose stem cells (ASCs) were isolated from the stromal vascular fraction (SVF) derived from abdominal adipose tissue biopsies from 5 different patients, as described elsewhere [23,24] and cultured for cell expansion in DMEM with 20% FBS, 2 mM L-glutamine, 100 U/mL penicillin, 100 μg/mL streptomycin and 1 μg/mL amphotericin-B. All the experiments where then performed in 10% FBS-DMEM. ASCs were used between passages 2–7. Primary cell populations had been obtained from subjects undergoing elective abdominal surgery, after signing written informed consent, according to the University Hospital Ethical Committee Approval (Ref.58/11, 15/10/2015). Human microvascular endothelial cells (HMEC-1), kindly obtained from the Centre for Disease Control and Prevention (Atlanta, GA, USA), were cultured in MCDB-131 medium (GIBCO-Fisher Scientific, Thermo Fisher Scientific, Waltham, MA, USA) supplemented with 10% HyClone^®^ fetal bovine serum (Thermo Fisher Scientific), 1 μg/mL hydrocortisone, 10 μg/mL epidermal growth factor, 2 mM L-glutamine, 100 U/mL penicillin, 100 μg/mL streptomycin. All cells were incubated at 37 °C in a humidified 5% CO_2_ atmosphere.

### 4.4. Co-Culture Experiments

#### 4.4.1. H295R-ASCs Co-Culture

H295R and ASCs were co-cultured using ThinCert^TM^ tissue culture inserts for 6-well plates with 0.4 µm pore size (Greiner Bio-One, Kremsmünster, Austria). H295R cells and ASCs were seeded in cell culture inserts (10^5^ cells/insert) and in wells (8 × 10^4^) respectively, unless otherwise indicated, with their own complete medium containing 10%FBS for 2 days before the co-culture experiments. At the co-culture starting time, inserts containing H295R were transferred into the wells containing ASCs, and all cells were grown in DMEM plus 10% FBS. H295R and ASCs cultured alone under the same conditions were used as controls for each assay.

#### 4.4.2. H295R Co-Culture with ASCs during In-Vitro Induced Adipogenesis

In order to differentiate towards mature adipocytes (ADIPO), ASCs were seeded onto glass coverslips in 6-well plates (5 × 10^4^ cells/well) and cultured alone or with H295R (10^5^ cells/insert), in the presence of adipogenic medium (10% FBS,-DMEM, 0.5 mM 3-isobutyl-1-methylxanthine, 1 µM dexamethasone, 10 µM insulin and 1 μM rosiglitazone–DIM cocktail) for 10 days to induce differentiation in ASCs. In parallel, ASCs cultured only in 10% FBS-DMEM (i.e., not adipogenic medium) for the same time interval were used as a negative control.

#### 4.4.3. H295R-mADIPO Co-Culture

ASCs plated in wells were induced to differentiate towards mature adipocytes (mADIPO) for 10 days with DIM cocktail (10% FBS-DMEM, 0.5 mM 3-isobutyl-1-methylxanthine, 1 µM dexa-methasone, 10 µM insulin and 1 μM rosiglitazone–DIM cocktail) and then cultured alone (mADIPO alone) or in the presence of H295R (mADIPO+H295R) for an additional 7 days in complete DMEM/F12 medium.

### 4.5. Cell Count

24 h-starved H295R cells and ASCs were co-cultured for 2, 3, 7 and 9 days, the cells were then trypsinized and counted using a hemocytometer. The mean number of cells was obtained by counting four replicates in at least three different experiments.

### 4.6. Glucose Uptake Measurement

ASCs and H295R, grown in mono- or co-culture for 7 or 9 days respectively, were washed twice with PBS and incubated overnight in serum free and low glucose (0.55 mM) medium. After rinsing in PBS, the cells were incubated in Hepes buffer (140 mM NaCl, 20 mM Hepes-Na pH 7.4, 2.5 mM MgSO_4_, 1 mM CaCl_2_, 5 mM KCl) containing 2-deoxy-[^3^H]-D-glucose [1 μCi/μL] for 10 min at 37 °C, washed with cold PBS, and lysed in 100 mM NaOH for 1 h at 37 °C. After resuspension of samples in Insta-Gel Plus cocktail (Perkin-Elmer, Waltham, MA, USA) radioactivity was measured with a scintillation beta counter.

### 4.7. SDS-PAGE and Western Blot Analysis

After mono- or co-culture, cells were lysed in RIPA buffer (20 mM Tris pH 7.4, 150 mM NaCl, 0.2 mM EDTA, 1 mM OVA, 0.5% Triton-100 in ddH_2_O) supplemented with 100× phosphatase inhibitor and 100× protease inhibitor. After protein measurement by Coomassie method, 30 µg proteins for each sample were separated by SDS–PAGE and transferred onto PVDF membranes (Immobilon, Merck-Millipore, Burlington, Massachusetts, USA). Each membrane was incubated overnight with primary antibodies at 4 °C and with peroxidase-secondary IgG at room temperature for 1.5 h. Image acquisition and densitometric analysis were performed with Quantity One software on a ChemiDoc XRS instrument (BIO-RAD Labs, CA, USA). All Western blots were repeated in at least three independent experiments. GAPDH or actin were used as internal loading control to normalize protein expression in each lane.

### 4.8. mRNA Isolation and Quantitative Real-Time RT-PCR (RT-qPCR)

mRNA isolation from H295R cells and ASCs, 7-day co-cultured or grown alone, was performed using the RNeasy Mini Kit (Qiagen, Hilden, Germany), according to manufacturer’s instructions. For each RNA sample, cDNA was obtained by reverse transcription PCR starting from 250 ng of RNA in 50 μL final volume reaction (Taqman RT-PCR kit; Applied Biosystems, Foster City, CA, USA) using the following cycling conditions: 10 min at 25 °C, 30 min at 48 °C, 3 min at 95 °C, hold 4 °C. Further quantitative real-time PCR was carried out using primers and probes for the following genes: BMI-1 (Hs00180411_m1), Nanog (Hs02387400_g1), OCT-4 (Hs00999634), Leptin (Hs00174877), AdipoQ (Hs00605917_m1), FABP4 (Hs00609791_m1), HSL (Hs00193510_m1), IL-8 (Hs00174103_m1), IGF2 (Hs04188276_m1), IGF-1R (Hs00181385_m1), Ob-R (HS001974497_m1), DPP4 (Hs00175210_m1), CXCR7 (Hs00604567_m1), CXCL12 (Hs00171022_m1), GAPDH (4352934) (#4331182,Taqman Gene Expression Assay; Applied Biosystems) and Taqman Universal Master mix (#4364338, Applied Biosystems). RT-PCR reactions, performed in triplicate for each gene, were carried out in 12.5 μL final volume on an ABI Prism 7900 Sequence Detector (Applied Biosystems) with the following cycling conditions: 15 s at 95 °C plus 1 min at 60 °C for 40 cycles. The amount of target genes, normalized to the endogenous reference gene (GAPDH) and related to a calibrator (Stratagene, San Diego, CA, USA), was calculated by 2^−ΔΔCt^ [60].

### 4.9. Glucose and Lactate Measurements

Conditioned media from mono- or co-cultures were collected, high–speed centrifuged and stored at −80 °C. Glucose and lactic acid were measured in frozen conditioned media from mono- and co-cultures by colorimetric assay (Siemens Healthcare, Tarrytown, NY, USA).

Intracellular lactate was measured using a specific Lactate Colorimetric/Fluorometric Assay Kit (Biovision, Milpitas, CA, USA), according to the manufacturer’s instructions. Briefly, cells grown for 7 days in mono- or co-culture conditions were collected and lysed in RIPA buffer. Thirty μL of cell lysates were added to a 96-well plate, diluted to 50 μL with the provided Lactate Assay Buffer, then mixed with 50 μL of the Reaction Mix and incubated at room temperature for 30 min. The absorbance values measured at 570 nm wavelength on a microplate reader (VICTOR multilabel plate reader; Perkin-Elmer) were corrected for the background and interpolated on a standard curve to obtain the lactate concentration, then normalized on the relative cell number.

### 4.10. Transmission Electron Microscopy

After 7-day mono- or co-culture, H295R and ASCs were trypsinized and centrifuged at 1200 rpm for 5 min. Pellets were fixed in cold 2.5% glutaraldehyde and 2% formaldehyde in 0.1 M sodium cacodylate buffer (pH 7.4) for 1 h at room temperature and post-fixed in cold 1% osmium tetroxide in 100 mM phosphate buffer (pH 7.4) for 1 h at room temperature. Pellets were dehydrated in graded acetone, passed through propylene oxide, and embedded in epoxy resin. Ultrathin sections were stained with gadolinium acetate and alkaline bismuth subnitrate and examined under a JEM1010 electron microscope (Jeol, Akishima, Tokyo, Japan) at 80 kV.

### 4.11. Intracellular Lipid Content Quantification

Intracellular lipid content in ASCs and adipocytes was measured by AdipoRed^TM^ assay (Cambrex, East Rutherford, NJ, USA) according to the manufacturer’s instructions. Fluorescence emission was measured by 485/572 nm excitation/emission. Specific absorbance related to differentiated adipocytes was calculated as fold increase on unspecific absorbance related to the undifferentiated ASCs. Coverslips were mounted on microscope slides in the presence of ProLong^®^ Gold antifade reagent with DAPI (Thermo Fisher Scientific) and fluorescence related to lipid droplets was acquired with a Leica DM4000 epifluorescence microscope (Leica Microsystems GmbH, Wetzlar, Germany).

### 4.12. Scratch Test

H295R cultured alone or in co-culture with ASCs for 7 days were trypsinized, seeded in a 12-well plate (1.5 × 10^6^ cells/well) and grown up to confluence. The confluent cell monolayer was scratched with sterile plastic 10 μL pipette, detached cells were eliminated by PBS wash and the medium was replaced, incubating cells for 48 h. In each well, migration was assessed under an inverted microscope and quantified using the MRI Wound Healing Tool of ImageJ software at designated time points 0, 24, 48 h post-scratch. The percentage of migration rate was expressed as: 100 × (1 − residual area)/(initial area).

### 4.13. Invasion Assay

Cell invasion assay was performed by a basement membrane-coated CytoSelect^TM^ 24-well cell invasion assay kit (Cell Biolabs, San Diego, CA, USA), according to the manufacturer’s instructions. Briefly, H295R cells previously cultured alone or co-cultured with ASCs for 7 days were trypsinized and plated in the upper chamber of the invasion plate (3 × 10^5^ cells/well) in serum-free culture medium. Complete culture medium (DMEM/F12 plus 10% FBS) was added to the lower chamber. Cells were incubated for 48 h at 37 °C in 5% CO_2_ atmosphere. Non-invasive cells were removed from the inside of the inserts that were then transferred to a clean well containing the Cell Stain Solution and incubated for 10 min at room temperature. Inserts were washed in water, allowed to air dry and transferred to an empty well containing the Extraction Solution. After 10 min incubation on an orbital shaker, samples were transferred to a 96-well microtiter plate and the OD 560 nm was measured in a microplate reader (VICTOR multilabel plate reader; Perkin-Elmer). The OD 560 nm value related to the Extraction Solution alone was used as the background subtraction factor. The assay was repeated for three different co-culture experiments.

### 4.14. Trans-Endothelial Migration Assay

HMEC-1 were cultured on 8 μm pore size inserts (CytoSelect™ tumor trans-endothelial migration assay kit, Cell Biolabs) (3 × 10^5^ cells/insert) for 72 h to allow monolayer formation, then activated with 1 ng/mL TNF-α for 1 h [61]. H295R, previously cultured alone or co-cultured with ASCs for 7 days, were trypsinized and incubated at 37 °C for 1 h with CytoTracker. The endothelial medium was removed and cancer cells were plated in the upper chamber of the invasion plate (3 × 10^5^ cells/well) without disturbing the endothelial monolayer. The insert was then transferred to another well containing the complete culture medium (DMEM/F12 plus 10% FBS). The assay was performed at 37 °C for 24 h. Non-migrating cells were removed, whereas those cells which had migrated to the bottom of the membrane were dissociated, lysed and quantified for fluorescence measurement at 480 nm/520 nm. Experiments were performed in three technical replicates and independently validated across two biological experiments.

### 4.15. F-actin Cytoskeleton Fluorescence Stain

H295R were seeded on glass coverslips in 6-well plates (10^5^ cells/well) and cultured alone or together with ASCs grown in cell culture inserts (8 × 10^3^ cells/insert). After 7 days, H295R were fixed in 4% PFA, permeabilized in PBS-0.2% Tryton and incubated with ActinGreen™ 488 ReadyProbes^®^ Reagent (Thermo Fisher Scientific) for 40 min to stain F-actin cytoskeleton. Coverslips were mounted on microscope slides in the presence of ProLong^®^ Gold antifade reagent with DAPI (Thermo Fisher Scientific). Fluorescence was acquired with a Leica DM4000 epifluorescence microscope (Leica Microsystems GmbH). Coverslips incubated with 5% horse serum-PBS were used as negative controls.

### 4.16. ELISA Assay

Conditioned media of ASCs cultured alone or with H295R for 7 days were analysed with an ELISA kit for human SDF-1a (CXCL12A) detection (Invitrogen, Thermo Fisher Scientific), according to the manufacturer’s instructions. Briefly, samples were incubated in the assay plate containing anti-SDF-1a antibody for 2.5 h at room temperature. After washing, samples were incubated with biotinylated antibody for 1 h at room temperature. Subsequent streptavidin-reagent and TMB substrate incubation were performed before measuring the OD 450/550 nm on a microplate reader (VICTOR multilabel plate reader; Perkin-Elmer). Each point was performed for a minimum of four times in at least three different experiments. Sample absorbance values were interpolated on a standard curve to obtain the relative concentrations. Assay sensitivity was 80 pg/mL. Each value was normalized on the relative cell number.

### 4.17. Statistical Analysis

Statistical analysis was performed using SPSS 22.0 software (SPSS Inc. Chicago, IL, USA). The Kolmogorov–Smirnov test was employed to assess the normal distribution of data, which were then expressed as mean ± SE. Student’s *t* test was applied for comparison of two classes of data as indicated in the figures. A *p* Value < 0.05 was considered as statistically significant.

## 5. Conclusions

Taken together, our results suggest that ASCs represent pivotal components of the tumor microenvironment, with a central role in mediating ACC metabolic and invasive reprogramming, by establishing a dynamic and bi-directional crosstalk with cancer cells and promoting tumor progression and metastasis. Further studies are needed to confirm the reciprocal role and weight of the molecular factors and pathways here identified as mediators of this cancer promoting interaction, with the perspective of identifying novel molecular targets more selectively druggable in invasive ACC.

## Figures and Tables

**Figure 1 cancers-11-01931-f001:**
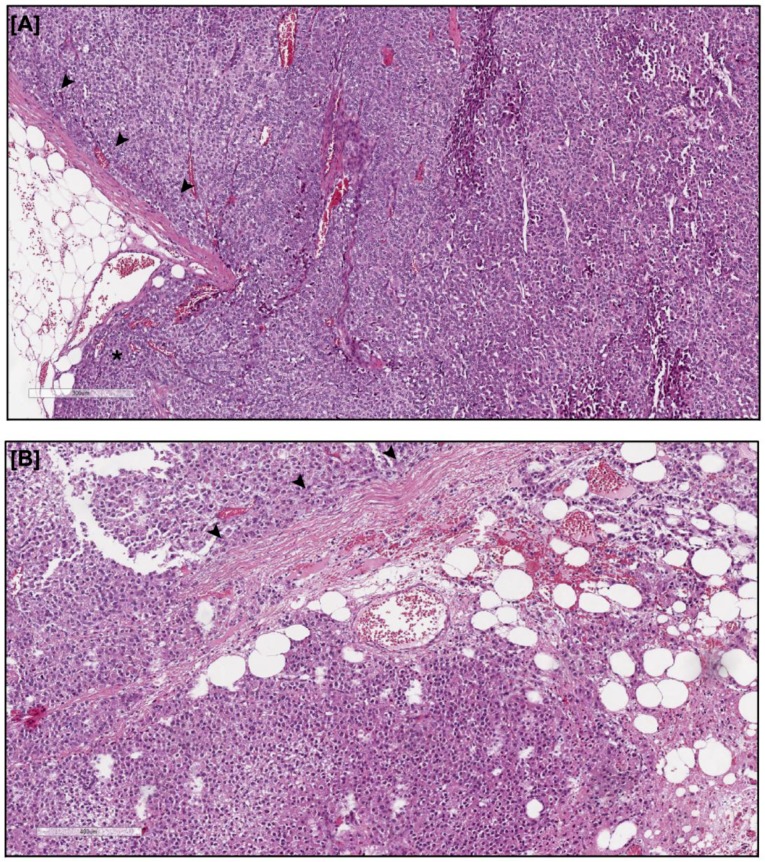
Capsular invasion in advanced ACC. (**A**) Representative Hematoxylin/Eosin staining of an advanced stage 3-ACC showing disruption of the capsule with pushing a well-circumscribed tumor border (*) into the surrounding adipose tissue. (**B**) Representative Hematoxylin/Eosin staining of an advanced stage 4-ACC displaying cancer extension beyond the capsule with irregular clusters and cords of tumor cells infiltrating the fat. Arrowheads indicate the remaining adrenal capsule. Scale bars = 300 μm (**A**) and 400 μm (**B**).

**Figure 2 cancers-11-01931-f002:**
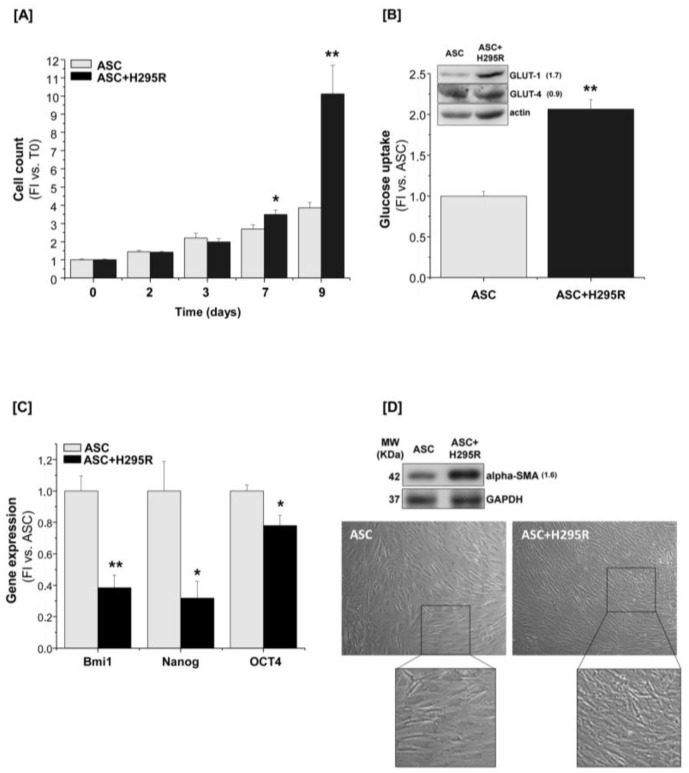
H295R cells stimulate ASC proliferation and drive ASC differentiation toward a myofibroblast-like phenotype. (**A**) ASCs alone (ASC) or co-cultured with H295R (ASC+H295R) were assessed for cell proliferation at the indicated time points (2, 3, 7 and 9 days) by direct cell count. The proliferative rate was calculated as fold increase (FI) versus the co-culture starting time (Time point = 0), *n* = 5. (**B**) Glucose uptake measurement and western blot analysis of GLUT-1 and GLUT-4 expression (inset, fold increase intensity vs. ASC after normalization on actin band is indicated to the right of the bands) assessed in ASCs after 7-day mono- or co-culture, *n* = 3. (**C**) Gene expression of specific mesenchymal stem-related markers revealed by RT-qPCR Taqman assay in 7-day co-cultured ASCs compared with the ASC mono-culture, *n* = 3. (**D**) Western blot analysis of α-SMA expression and optical microscopy of ASCs cultured alone or in the presence of H295R cells for 7 days. Original magnification: 10×; zoom in: 2×. For western blot analysis, GAPDH or actin were used as internal loading control. Gene expression and glucose uptake are indicated as fold increase (FI) versus ASCs alone. Data are expressed as the mean ± SE in at least three independent experiments; * *p* < 0.05; ** *p* < 0.001. Details of western blot can be viewed at the Appendix A.

**Figure 3 cancers-11-01931-f003:**
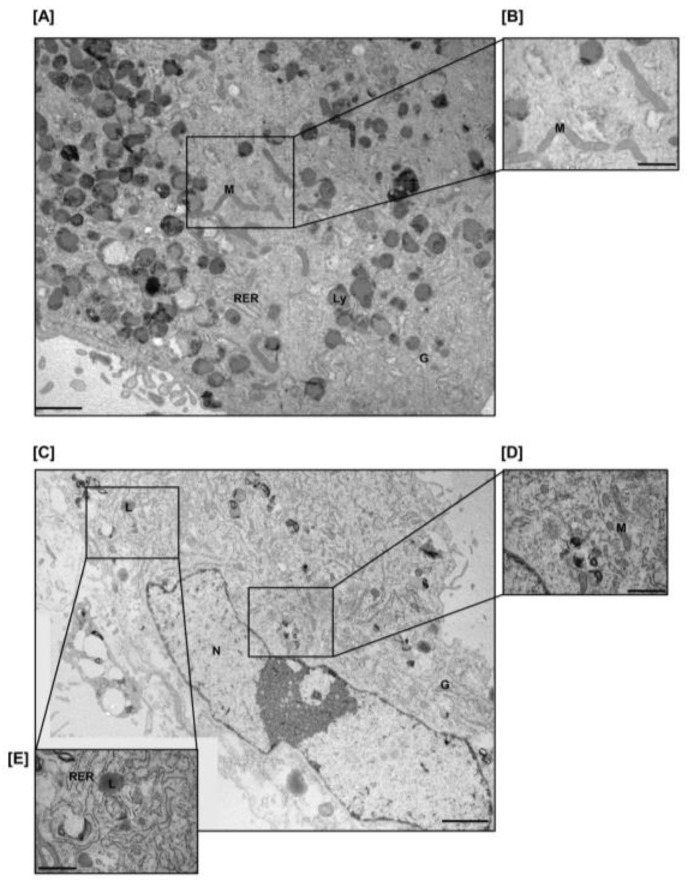
ASC ultrastructure shows radical changes after co-culture. ASCs cultured alone (**A**, zoom in **B**) or in the presence of H295R (**C**, zoom in: **D**,**E**) for 7 days, *n* = 3 independent experiment. Representative different fields with the same original magnification were digitally merged to reconstruct a wider cell portion. N = nucleus; M = mitochondria; RER = rough endoplasmic reticulum; Ly = lysosome; G = Golgi cisternae; L = lipid droplet. Scale bars = 2 μm (**A**,**C**) and 1 μm (**B**,**D**,**E**).

**Figure 4 cancers-11-01931-f004:**
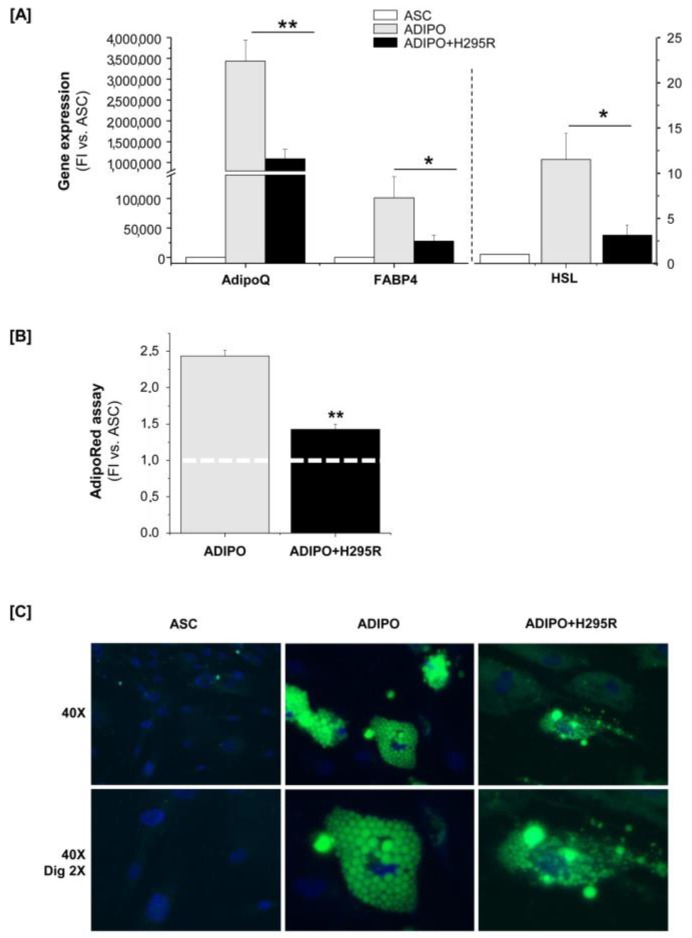
H295R cells affect ASC ability to efficiently differentiate toward mature white adipocytes. (**A**) ADIPOQ, FABP4 and HSL gene expression measured by RT-qPCR Taqman assay in undifferentiated ASCs (used as negative control) and adipocytes differentiated in vitro from ASCs alone (ADIPO) or induced to differentiate in co-culture with H295R cells (ADIPO+H295R) for 10 days. Gene expression was expressed as fold increase (FI) versus ASCs. The left Y axis refers to AdipoQ and FABP4 expression, the right one to HSL expression. (**B**) Adipocyte lipid content in ADIPO and ADIPO+H295R was calculated as fold increase (FI) versus undifferentiated ASCs (reference FI = 1, indicated by the white dashed line). (**C**) AdipoRed-related fluorescence stain of intracellular lipid droplets (green) in ASCs, ADIPO and ADIPO+H295R. Cell nuclei were counterstained with DAPI (blue). Original magnification is indicated for each row. Quantitative data are expressed as the mean ± SE in *n* = 4 independent experiments; * *p* < 0.05; ** *p* < 0.001.

**Figure 5 cancers-11-01931-f005:**
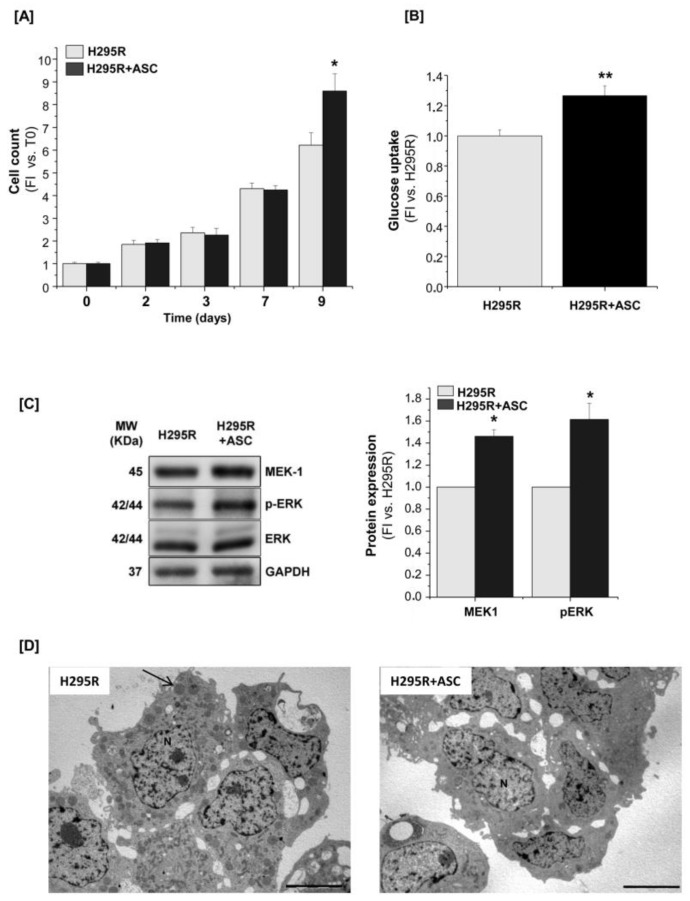
H295R cell proliferation rate increases in the presence of ASCs. (**A**) H295R cells alone (H295R) or co-cultured with ASCs (H295R+ASC) were assessed for cell proliferation at the indicated time points (2, 3, 7 and 9 days) by direct cell count. The proliferative rate was indicated as fold increase (FI) versus the co-culture starting time (T0: Time = 0). Glucose uptake measurement (**B**) and western blot analysis of MEK1, ERK and phospho-ERK expression (**C**) were compared in H295R cells after mono- or co-culture. For western blot analysis, GAPDH was used as internal loading control. Both glucose up-take and protein expression were calculated as fold increase (FI) versus H295R cells alone. (**D**) Electron microscopy of H295R cells cultured alone or together with ASCs for 7 days. The black arrow indicates the structures corresponding to lipid droplets. N = nucleus. Scale bars = 5 μm. Data are expressed as the mean ± SE vs. H295R cell alone, *n* = 3 independent experiments; * *p* < 0.05; ** *p* < 0.001. Details of western blot can be viewed at the Appendix A.

**Figure 6 cancers-11-01931-f006:**
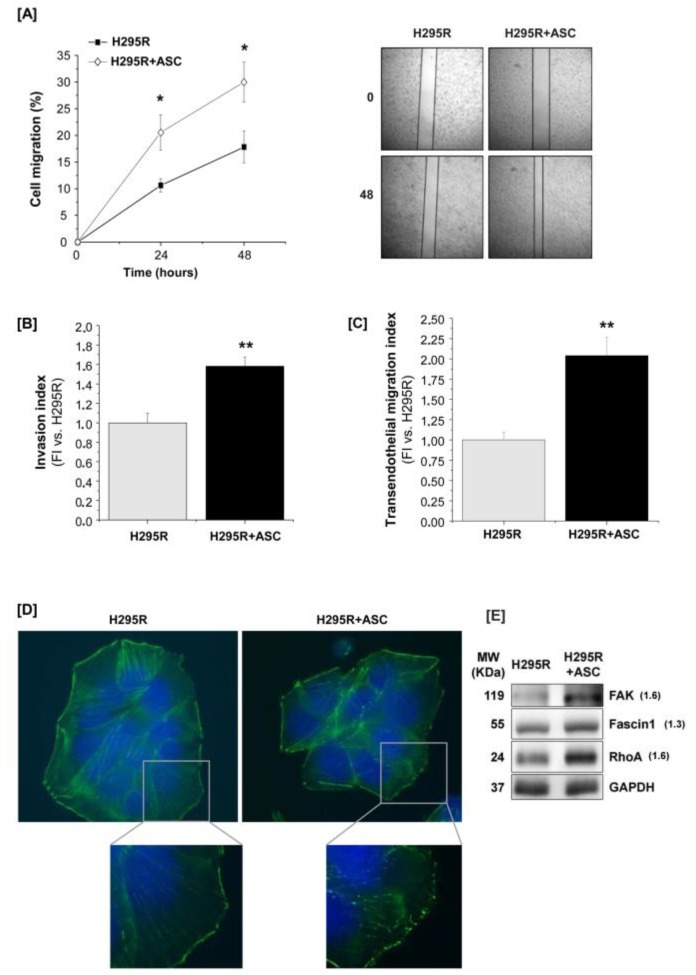
H295R cell migration and invasion ability is increased by ASC co-culture. (**A**) Scratch assay: cell migration at 0, 24 and 48 h post-scratch in H295R cells previously cultured alone (H295R) or co-cultured with ASCs (H295R + ASC) for 7 days, and related representative images of *n* = 3 independent experiments. Original magnification: 5×. (**B**) Cell invasion assay: the invasion index was calculated as fold increase (FI) versus H295R cells alone. (**C**) Trans-endothelial migration assay: the invasion index was calculated as fold increase (FI) versus H295R cells alone. (**D**) Fluorescence staining of F-actin cytoskeleton (in green) in H295R after 7-day mono- or co-culture. DAPI staining (in blue) was used to visualize cell nuclei. Original magnification: 100×; zoom in: 2×. (**E**) Western blot analysis of FAK, RhoA and Fascin-1 performed on H295R cells cultured alone or together with ASCs for 7 days. GAPDH was used as internal loading control (fold increase intensity vs. H295R cells after normalization on GAPDH band is indicated to the right of the bands). Quantitative data are expressed as the mean ± SE vs. H295R alone in at least three independent experiments; * *p* < 0.05; ** *p* < 0.001. Details of western blot can be viewed at the Appendix A.

**Figure 7 cancers-11-01931-f007:**
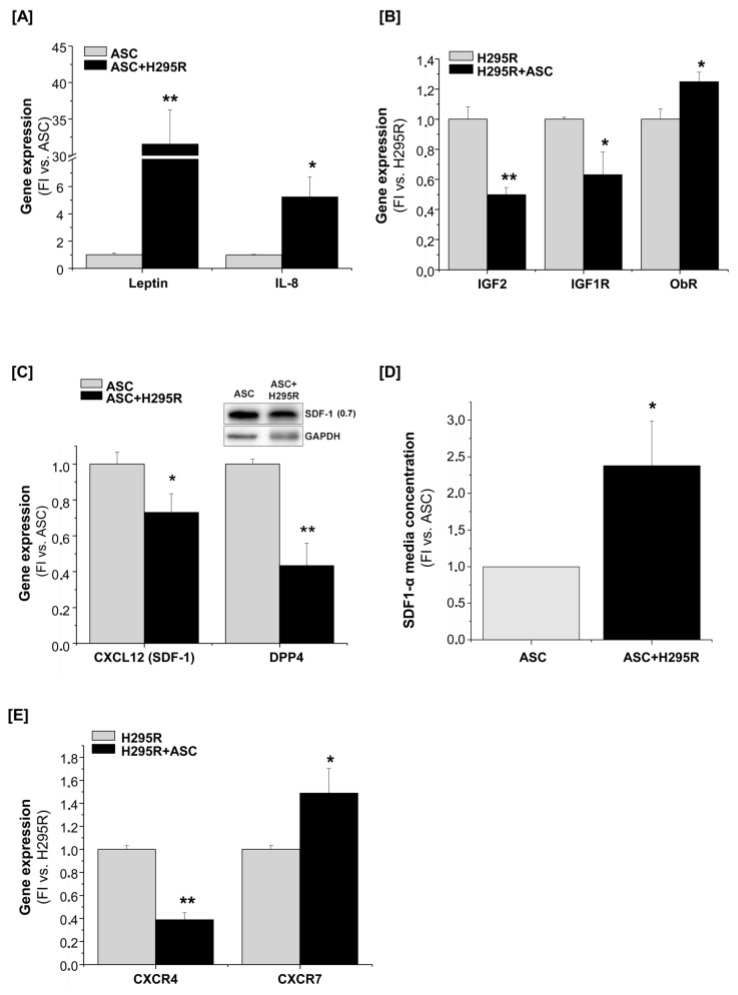
The potential molecular crosstalk underlying H295R/ASCs interaction**.** RT-qPCR Taqman analysis of gene expression in both ASCs and H295R cells after 7-day mono- or co-culture (ASC or H295R cells and ASC + H295R or H295R + ASC, respectively). Leptin and IL-8 (**A**), and CXCL12 (SDF-1) and DDP4 (**C**) gene expression was evaluated in ASCs, and the intracellular SDF-1 protein expression was assessed by western blot analysis (**C**), inset; fold increase intensity vs. ASCs after normalization on GAPDH band is indicated to the right of the band). IGF2, IGF-1R and Ob-R gene expression was evaluated in H295R cells (**B**). For all genes, the relative expression level was calculated as fold increase (FI) versus the mono-culture. For western blot analysis, GAPDH was used as internal loading control. (**D**) Extracellular levels of SDF-1 measured by ELISA assay in the conditioned medium of ASCs after 7-day mono- and co-culture. The absolute SDF-1 concentrations were normalized on the relative cell number and expressed as fold increase (FI) versus ASCs. (**E**) CXCR4 and CXCR7 expression was evaluated in H295R cells alone or in co-culture with ASCs. Data are expressed as the mean ± SE vs. the respective monoculture in at least three independent experiments; * *p* < 0.05; ** *p* < 0.001 Details of western blot can be viewed at the Appendix A.

**Figure 8 cancers-11-01931-f008:**
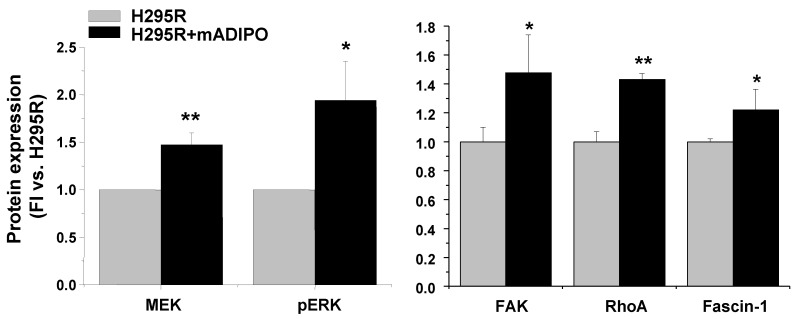
Expression of proliferation- and migration-related proteins in H295R cells co-cultured with mature adipocytes**.** Protein expression was assessed by western blot analysis in cell samples in H295R cells co-cultured for 10 days with previously in vitro differentiated adipocytes (H295R+mADIPO) compared with H295R mono-culture. Bar charts represent protein expression quantified by densitometric analysis of protein bands normalized on GAPDH, used as internal loading control, and calculated as fold increase vs. H295R cells. Data are expressed as the mean ± SE in n = 3 independent experiments, * *p* < 0.05, ** *p* < 0.001 vs. H295R cells alone.

**Figure 9 cancers-11-01931-f009:**
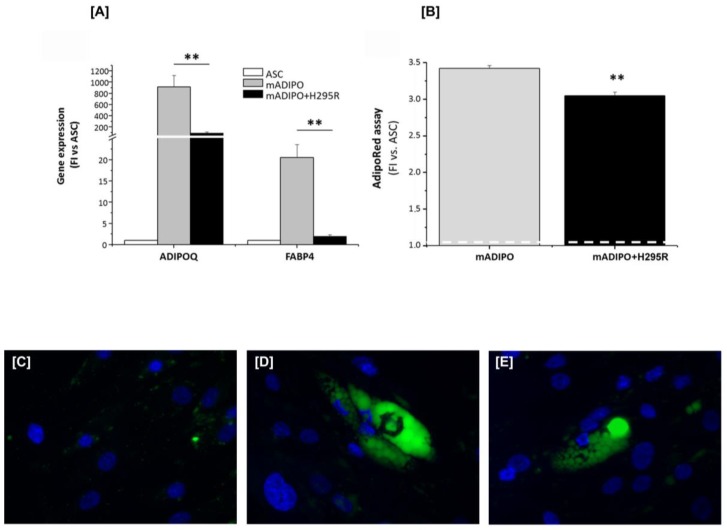
Co-culture of H295R cells with mature adipocytes resulted in de-differentiation of the adipose cells. (**A**) RT-qPCR Taqman assay was performed on cell samples from undifferentiated ASCs, used as negative control, and in vitro differentiated adipocytes cultured alone (mADIPO) or co-cultured with H295R cells (mADIPO+H295R) for 7 days. Gene expression related to AdipoQ and FABP4 is expressed as fold increase (FI) vs. undifferentiated ASCs used as control. (**B**) Adipocyte lipid content, assessed by AdipoRed assay, was calculated as fold increase (FI) vs. undifferentiated ASCs, (indicated by the white dashed line as = 1). Data are expressed as the mean ± SE in *n* = 3 independent experiments, ** *p* < 0.001 vs. mADIPO alone. Representative images of AdipoRed-related fluorescence staining intracellular lipid droplets (green) in samples from undifferentiated ASCs (negative control, (**C**)), mADIPO (**D**) and mADIPO+H295R (**E**). Cell nuclei were counterstained with DAPI (blue). Original magnification: 40×.

**Table 1 cancers-11-01931-t001:** Glucose and lactic acid changes in co-culture.

Metabolite	ASC	ASC + H295R	H295R	H295R + ASC
Conditioned medium				
*Glucose*	1.0 ± 0.02	0.67 ± 0.03 **	1.0 ± 0.05	0.83 ± 0.02 *
*Lactic acid*	1.0 ± 0.10	1.96 ± 0.25 *	1.0 ± 0.02	1.70 ± 0.05 **
Intracellular lactate				
	1.0 ± 0.02	1.39 ± 0.03 **	1.0 ± 0.10	1.33 ± 0.11 *

Glucose and lactic acid concentrations measured in the conditioned medium of ASCs and H295R cells in mono-or co-culture and the related intracellular lactate content after 7 days of co-culture. The absolute concentration was normalized on the cell number and expressed as fold increase versus the relative mono-culture. Data are expressed as the mean ± SE in at least three independent experiments; * *p* < 0.05, ** *p* < 0.001 vs. the respective mono-culture.

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
