# Peer review of "The Adipose Stem Cell as a Novel Metabolic Actor in Adrenocortical Carcinoma Progression: Evidence from an In Vitro Tumor Microenvironment Crosstalk Model"

_cancers, 2019, doi:10.3390/cancers11121931_

Round 1

Reviewer 1 Report

This study on the role of ASC in modulation of cancer phenotype is overall well designed and presented. Here are few areas on which it can be improved.

Although the authors have previously published in similar areas, adequate characterization of isolated ASC cells remain important for the interpretation of results. Those data should be included in supplement.  All experiments were performed with only one ACC cell line H295R. Efforts should be made to study at least one other newly published cell line. H295Rs are in fact unique in harboring a p53 mutation along with beta catenin activation. Since relation of glucose metabolism is now well tied to beta catenin activation, some of the effects observed might be cell specific and not necessarily translatable.   Also none of the immunoblots have any quantification.  It is not enough to show to only one advanced ACC with capsular invasion. The authors should attempt to show few more samples to justify the importance of the study.

Author Response

REVIEWER 1

We thank the Reviewer for the careful reading of our manuscript and for the constructive comments made. Accordingly, we added new data to the manuscript (two additional figures for mature adipocyte effects) and made changes that are highlighted throughout the revised version of the manuscript. Please note that due to the amount of work done for the revision of the manuscript and addition of novel data and experiments, an equal first co-authorship was attributed to Roberta Armignacco and Giulia Cantini. We hope that we satisfy the comments raised and that this version of the manuscript may be found acceptable for publication in Cancers.

The point-by-point-reply follows.

This study on the role of ASC in modulation of cancer phenotype is overall well designed and presented. Here are few areas on which it can be improved.

Q.1.Although the authors have previously published in similar areas, adequate characterization of isolated ASC cells remain important for the interpretation of results. Those data should be included in supplement.  

R.1. ASC characterization has been performed in all the cell populations used in this study by assessing their adipogenic potential when in vitro-induced towards differentiation (according to the method described in the manuscript). For all populations (derived from n=5 different subjects) the quantitative expression of AdipoQ, FABP4 and HSL was evaluated, as markers of adipogenesis. Moreover, intracellular triglyceride accumulation in lipid droplets was evaluated by AdipoRed staining followed by cell visualization by epifluorescence microscopy. These data are shown in Figure 4A (AdipoQ, FABP4 and HSL quantitative gene expression) and Figure 4C (AdipoRed fluorescence microscopy), where ASCs are the undifferentiated cells and ADIPO correspond to the in vitro-induced adipocytes.

Q.2.All experiments were performed with only one ACC cell line H295R. Efforts should be made to study at least one other newly published cell line. H295Rs are in fact unique in harboring a p53 mutation along with beta catenin activation. Since relation of glucose metabolism is now well tied to beta catenin activation, some of the effects observed might be cell specific and not necessarily translatable.  

R.2.We agree with the Reviewer’s comment that this might be possible. However, so far, H295R cells are the only one commercially available cell line considered a valid model of adrenocortical carcinoma. In fact, SW13 are derived from an adrenal metastasis of lung cancer, and Muc1 cell line, recently developed in Munich’s lab, are promising but still currently under characterization.

Q.3. Also none of the immunoblots have any quantification.  

R.3. Quantification of band intensity relative to the internal control loading protein (GAPDH or actin) has now been added for all blots, in addition to the one already present in Fig. 5C.

Q.4. It is not enough to show to only one advanced ACC with capsular invasion. The authors should attempt to show few more samples to justify the importance of the study.

R.4. An additional image from a patient with advanced stage 4-ACC, showing cancer infiltration of the adjacent adipose tissue have now been added to Fig.1 (panels B). Moreover, we have now added the percentage of capsular invasion (one of the parameter taken into account in the Weiss score analysis of ACC, which corresponds to the disruption of the adrenal capsule and tumor infiltration of the surrounding adipose tissue) in our cohort of advanced ACC (stage 3-4), to show the relevance of the phenomenon (89%, 17/19).

Reviewer 2 Report

Authors present results of the co-culture model of adipose stem cells with adrenal cancer cells indicating an important crosstalk between those two compartments.

The biggest problem of the study is the lack of appropriate control. Authors should consider adding another cancer cell line which normally has no contact with adipose tissue to further assess if those effects are truly relevant and specific for adrenal cancer microenvironment. It could be also interesting to assess whether co-culture with ASC changes the susceptibility to immune-cell mediated cytotoxicity (e.g. by gamma-delta T cells or iNKT cells, the latter being highly abundant in adipose tissue).

Authors assume that it is IL-8 and leptin that are responsible for the observed effect. It would be highly useful to specifically block those proteins by using specific mAbs and then assessing the effect of co-culture. Also, what is the effect of monoculture of adrenal cancer cells with the addition of IL-8 and leptin? Another important step would be to actually assess the co-culture model with real-patient samples, probably with adipose cells at different stages of differentiation.

Please provide the number of experiments in each figure. Please provide clones of antibodies and/or the catalogue number for ALL Abs used. Please provide data about secondary antibodies - which secondary Abs were used with which primary Abs. For how long prior to the study were the cell lines cultured? Did the authors check if they are phenotypically stable overtime? What primary cell populations were derived from patients? Please provide catalogue numbers for all TaqMan probes. Please use the correct term for the method - authors had performed neither RT-PCR nor qRT-PCR, but RT-qPCR. Please correct where necessary. Please provide the name and catalogue number for the master mix used. What was the reference for 2-ddCt? Please correct the sentence in line 66 and 129-131.

Author Response

REVIEWER 2

We thank the Reviewer for the careful reading of our manuscript and for the constructive comments made. Accordingly, we added new data to the manuscript (two additional figures for mature adipocyte effects) and made changes that are highlighted throughout the revised version of the manuscript. Please note that due to the amount of work done for the revision of the manuscript and addition of novel data and experiments, an equal first co-authorship was attributed to Roberta Armignacco and Giulia Cantini. We hope that we satisfy the comments raised and that this version of the manuscript may be found acceptable for publication in Cancers.

The point-by-point-reply follows.

Authors present results of the co-culture model of adipose stem cells with adrenal cancer cells indicating an important crosstalk between those two compartments.

Q.1.The biggest problem of the study is the lack of appropriate control. Authors should consider adding another cancer cell line which normally has no contact with adipose tissue to further assess if those effects are truly relevant and specific for adrenal cancer microenvironment.

R.1.The concept of an “appropriate control” for the effects exerted by the adipose microenvironment on ACC is quite difficult to identify. In fact, as we hope to have made clear in the Introduction and in the Discussion sections, the adipose microenvironment has also been described to exert similar effects supporting tumor progression in other solid tumors, where the tumor mass is physiologically in close contact with the fat mass. Moreover, also the relative effects exerted by the tumor mass on the adipose cells (adipose precursors and mature adipocytes) are common to other solid tumors (see references 4, 9-13, 32). Thus, these effects are not exclusive of the adrenocortical cancer. Assessing the effects exerted by adipose cells on tumors that are not physiologically in contact with fat is rather tricky as it is an artificial situation, whatever the result is.

Thus, in order to show the Reviewer our effort to deeply understand the complex interaction occurring between the adipose microenvironment and cancer in the adrenal organ, we performed the same type of ASC co-culture experiments set up for H295R cells (corresponding to the steroidogenic cortical region of the adrenal), also with MPC cells obtained from pheochromocytoma (a neuroendocrine tumor of the adrenal medulla). We chose this tumor as it is representative of the neuroendocrine component of the adrenal organ, and in addition should not be in direct contact with the adrenal associated fat. We observed a significant reduction in the number of MPC cancer cells when co-cultured with ASCs in the same conditions used in this study; fold decrease vs. MPC alone: FI2 days = 0.71  (p<0.0001), FI3 days = 0.71  (p<0.0001),  FI7 days =0.90 (p<0.05),  FI9 days = 0.62 (p<0.0001). 

These findings suggest a difference in the effects exerted by ASC co-culture with different adrenal tumors: promoting cell growth in the adrenocortical cancer and decreasing it in the medullary tumor. These data are not published and they are a part of a different study that we are performing on pheochromocytoma models.

Q.2.It could be also interesting to assess whether co-culture with ASC changes the susceptibility to immune-cell mediated cytotoxicity (e.g. by gamma-delta T cells or iNKT cells, the latter being highly abundant in adipose tissue).

R.2. We thank the Reviewer for this comment, as the immune system component of the tumor microenvironment and its interaction with the cancer cells are interesting aspects of the complex cancer microenvironment for their translational potential targets for innovative immunotherapies against cancer. Indeed, we are currently setting up a complex model for studying the interactions occurring between immune infiltrating cells (macrophages, lymphocytes and monocytes, iNKT cells) and ASC-H295R co-cultures, to study how the presence of ASCs can modulate the immune response of adrenocortical cancer in this microenvironment immune context. However, this interesting aspect goes beyond the aim of the present study, which was focused on assessing the potential interaction between ASCs- and H295R cells in a dual system.  

Q.3.Authors assume that it is IL-8 and leptin that are responsible for the observed effect. It would be highly useful to specifically block those proteins by using specific mAbs and then assessing the effect of co-culture. Also, what is the effect of monoculture of adrenal cancer cells with the addition of IL-8 and leptin?

R.3. We agree with the Reviewer that this aspect is very intriguing. However, the goal of this study was to assess the reciprocal effects of ASCs and H295R cells in a co-culture system. We described the effects on cell proliferation, metabolism, migration and invasion. In particular, we found that when in co-culture with ASCs, H295R cells undergo a shift in the paracrine control of their proliferation, regulated by an endogenous production of IGF2, towards factors controlled by ASCs, such as IL-8, leptin or SDF1. This shift in the cancer regulating axes deserves a dedicated study, which we are now performing, and that includes the experiments suggested by the Reviewer not only for IL-8 and leptin, but also for SDF1, to assess the reciprocal weight of these axes also in the immunomodulatory context of ASC-H295R cell interaction (see the reply to the previous comment).   

Q.4.Another important step would be to actually assess the co-culture model with real-patient samples, probably with adipose cells at different stages of differentiation.

R.4. Since it is not possible to investigate in vivo or ex vivo in the ACC patients the interaction occurring in vitro between the adipose microenvironment and H295R characterized by our experimental design, we have now added to the revised manuscript the results  also obtained by co-culture H295R cells with mature adipocytes and not only with the adipose precursors. Indeed, in the adipose tissue in vivo, the mature adipocyte component is overwhelming respect to undifferentiated stem stages. Therefore, the interaction with mature adipocytes may better reflect the physiological conditions.

We have now added new experiments showing the inhibitory effects exerted on H295R cell signaling of MEK, pERK, FAK, RhoA, fascin-1 (new figure 8) and on the adipose differentiation markers (AdipoQ and FABP4, new figure 9A) and intracellular lipid accumulation (new figure 9 B-E), when cancer cells are co-cultured with mature adipocytes, similarly to what it was obtained when co-culture was performed with ASCs.

Q.5.Please provide the number of experiments in each figure.

R.5. Done.

Q.6.Please provide clones of antibodies and/or the catalogue number for ALL Abs used. Please provide data about secondary antibodies - which secondary Abs were used with which primary Abs.

R.6. Done.

Q.7. For how long prior to the study were the cell lines cultured? Did the authors check if they are phenotypically stable overtime? What primary cell populations were derived from patients?

R.7. We have previously isolated and characterized primary populations of Adipose-derived Stem Cells (ASCs) from human adipose tissue specimens, as described in Baglioni et al., FASEB J 2009; Baglioni et al., PLOS 2012. For this study, experiments have been carried out in primary cell populations derived from 5 different subjects. A sentence has been added in the paragraph dedicated to Cell Culture. Ethical approval to the study for ASC isolation from adipose tissue specimens has been added (lines 505-508). All information requested has been added in the revised version of the manuscript.

Q.8.Please provide catalogue numbers for all TaqMan probes.

R.8.Done.

Q.9.Please use the correct term for the method - authors had performed neither RT-PCR nor qRT-PCR, but RT-qPCR. Please correct where necessary.

R.9.Done

Q.10.Please provide the name and catalogue number for the master mix used.

R.10.Done.

Q.11.What was the reference for 2-ddCt?

R.11. The amount of expression of the target gene was firstly normalized to the expression of the GAPDH reference gene in each sample and then compared to a calibrator. A sentence is present in the Method section (lines 581-583): “The amount of target genes, normalized to the endogenous reference gene (GAPDH) and related to a calibrator (Stratagene, San Diego, CA, USA), was calculated by 2-ΔΔCt.”

The requested reference has now been added.

Q.12.Please correct the sentence in line 66 and 129-131.

R.12. Done.